# Social media does not elicit a physiological stress response as measured by heart rate and salivary cortisol over 20-minute sessions of cell phone use

**Suzanne Oppenheimer**[1]*, **Laura Bond**[2], **Charity Smith**[1]¤

**1** Department of Biological Sciences, College of Western Idaho, Boise, Idaho, United States of America,
**2** Biomolecular Research Center, Boise State University, Boise, Idaho, United States of America

¤ Current address: TRIO Academic Coaching & Education Support, Boise State University, Boise, Idaho, United States of America

* suzanneoppenheimer@cwi.edu

## Abstract

The pervasive use of social media has raised concerns about its potential detrimental effects on physical and mental health. Others have demonstrated a relationship between social media use and anxiety, depression, and psychosocial stress. In light of these studies, we examined physiological indicators of stress (heart rate to measure autonomic nervous system activation and cortisol to assess activity of the hypothalamic-pituitary-adrenal axis) associated with social media use and investigated possible moderating influences of sex, age, and psychological parameters. We collected physiological data from 59 subjects ranging in age from 13 to 55 across two cell phone treatments: social media use and a pre-selected YouTube playlist. Heart rate was measured using arm-band heart rate monitors before and during cell phone treatments, and saliva was collected for later cortisol analysis (by enzyme immunoassay) before and after each of the two cell phone treatments. To disentangle the effects of cell phone treatment from order of treatment, we used a crossover design in which participants were randomized to treatment order. Our study uncovered a significant period effect suggesting that both heart rate and cortisol decreased over the duration of our experiment, irrespective of the type of cell phone activity or the order of treatments. There was no indication that age, sex, habits of social media use, or psychometric parameters moderated the physiological response to cell phone activities. Our data suggest that 20-minute bouts of social media use or YouTube viewing do not elicit a physiological stress response.

## Introduction

With widespread use of individually-accessible screens (e.g., smart phones, tablets, and computers), many people have expressed concern over excessive use; for example, the American Academy of Pediatrics [1], Britain's Chief Medical Officer [2], and eminent psychologist Jean

**Data Availability Statement:** Data for this study are available at: https://doi.org/10.18122/brc_data.1.boisestate.

**Funding:** SO,LB,CS: This research was made possible by Institutional Development Awards (IDeA) from the National Institute of General Medical Sciences of the National Institutes of Health under Grant #P20GM103408. LB: Additional support was offered by Grants # 1U54GM104944, P20GM109095, and 1C06RR020533. We also acknowledge support from The Biomolecular Research Center at Boise State, BSU-Biomolecular Research Center, RRID:SCR_019174, with funding from the National Science Foundation, Grants #0619793 and #0923535; the M. J. Murdock Charitable Trust; Lori and Duane Stueckle, and the Idaho State Board of Education. The funders had no role in study design, data collection and analysis, decision to publish, or preparation of the manuscript.

**Competing interests:** The authors have declared that no competing interests exist.

Twenge [3] have issued warnings against overuse of smartphones by children. Problematic smart-phone use is most common in adolescents [4], but there is also evidence that excessive smart-phone use has negative repercussions across all age groups [5]. While screens are the instrument, social media is one of the elements often implicated in the dangers associated with screen time. Unlike in-person social interactions, which are often associated with higher psychological well-being [6,7], excessive social media interactions have been linked to poor psychological well-being [8–10].

Some studies have suggested that social media use may influence psychosocial stress [11–13], and social media use has frequently been linked to conditions such as anxiety and depression [14–16]. Fear of missing out, the worry that one may miss out on important interpersonal interactions or events [17], seems to both drive excessive use of social media and contribute to depression and anxiety associated with its use [18,19]. In some cases, social media elicits low self-esteem, poor body image, and envy [20,21]; such feelings are thought to be evoked when social media users evaluate their social and personal worth by comparing themselves to others who have curated and/or enhanced an idealistic online image [22]. Poor sleep quality, which has been linked to computer, internet, and social media use [16], is also thought to contribute to depression, anxiety, and low self-esteem [23,24]. Taken together, the body of literature suggests that multiple factors related to social media contribute to its association with psychosocial stress. However, it is unclear whether social media use leads to stress or if stress leads to social media use. This study attempts to fill this gap in the literature by addressing whether short bouts of social media use evoke stress.

Psychological stress is known to stimulate a physiological stress response, mediated by the endocrine and autonomic nervous systems [25–27]. Stress activates the hypothalamic-pituitary-adrenal (HPA) axis and the sympathetic division of the autonomic nervous system, and salivary cortisol and heart rate are commonly measured biomarkers of this activity [28,29]. While the stress response helps maintain physiologic balance in the face of sudden physiological or psychological perturbations [30], chronic activation of this response can result in an increased allostatic load and disease [31]; for example, stress has been linked to depression, cardiovascular disease, and type 2 diabetes [32–34, respectively]. If the use of social media causes psychosocial stress and subsequent activation of the physiological stress response, then protracted use could have deleterious consequences for both psychological and physical health. Several studies have examined the relationship between physiological variables (e.g., cardiovascular measures and cortisol) and social media use, but the findings are not consistent (Table 1). This points to a more nuanced relationship between social media use and physiological and psychological stress; we feel that there is a need for further research in this area to help tease apart this relationship.

Unlike these studies (Table 1) that examine correlations or social media's influence on an acute social stressor (i.e. TSST), the focus of our controlled study is to evaluate the physiological response to a short-term bout of social media use and indirectly take into account the potential moderating effects of age, sex, and psychosocial factors. We hypothesized that social media use would impart some degree of psychosocial stress that would activate the physiological stress response more so than our control treatment (non-evocative You-tube playlist). To assess the physiological response, we measured salivary cortisol to quantify activity of the HPA axis and heart rate to assess activation of the sympathetic nervous system. We examine the physiological effects of short bouts of social media because this is often how it is accessed (e.g., as we wait for a friend or take a break from another task). If this type of usage results in a stress response, then this could be one potential mechanism by which social media poses a health risk.

**Table 1. Summary of key findings in studies examining the relationship between physiological variables and social media.**

| Physiological Variables | Key Findings | Reference |
|---|---|---|
| Cortisol | diurnal cortisol levels positively associated with number of Facebook friends and negatively associated with Facebook peer interactions | [35] |
| Cortisol | Facebook vs. control treatment following Trier Social Stress Test (TSST): -greater sustained cortisol following Facebook use | [13] |
| Cortisol | adolescents who report more frequent use of social media have a greater cortisol awakening response | [11] |
| cortisol heart rate (HR) diastolic/systolic blood pressure (BP) | Facebook vs. control treatment followed by Trier Social Stress Test (TSST): -lower increase in systolic BP in Facebook than control -no difference in HR recovery, diastolic BP, or salivary cortisol | [36] |
| mean arterial pressure (MAP) heart rate | social media use following TSST facilitated recovery of MAP & HR | [37] |
| Cortisol | salivary cortisol positively associated with social media usage and addiction | [38] |

HR = heart rate, BP = blood pressure, MAP = mean arterial pressure.

## Material and methods

### Ethics statement

The study protocol was approved by the Institutional Review Board of The College of Idaho (1819-CWI00039.3). Adult subjects signed written informed consent forms, and those under the age of 18 years old obtained written consent from a parent, guardian, or legally authorized representative. Participants self-reported their biological sex and age on surveys.

### Participants & overall procedures

Participants were recruited in July 2019 using fliers, internet announcements, and direct oral solicitation, primarily directed toward faculty, students, and staff at the College of Western Idaho. The study was broadly open to adults, but individuals under 18 years were encouraged to join with parental consent. Because cortisol levels are elevated during pregnancy [39], pregnant women were excluded from the study. Those with Cushing's syndrome were also excluded because of pathologically elevated cortisol levels [40]. Subjects included in cortisol analyses had not consumed alcohol within 12 hours and had not eaten a meal within 60 minutes (identified through questionnaires). Because our cortisol assay exhibited low cross-reactivity ($\leq$0.015%) with steroids commonly used in hormonal contraceptives [41], women on birth control were not excluded from our analyses (12.5% of females reported the use of hormonal contraceptives).

Participants included 40 women and 19 men, ranging in age from 13 to 55 with a mean of 25.8 years and a standard deviation of 11.5 years. Twelve participants were under 18 (5 women and 7 men). The study was conducted over three days, with 15 participants on the first day, 21 on the second day, and 23 on the third day. On each of the three days, all participated simultaneously and were randomly assigned to each experimental group order. Heart rate data were not available for six participants. The cortisol data from three participants were not used in the analysis because they ate within the hour prior to the study. Cortisol values from two individuals at a single time point were unavailable for analysis due to processing issues. Otherwise, all individuals completed the entire study, including completing all questionnaires.

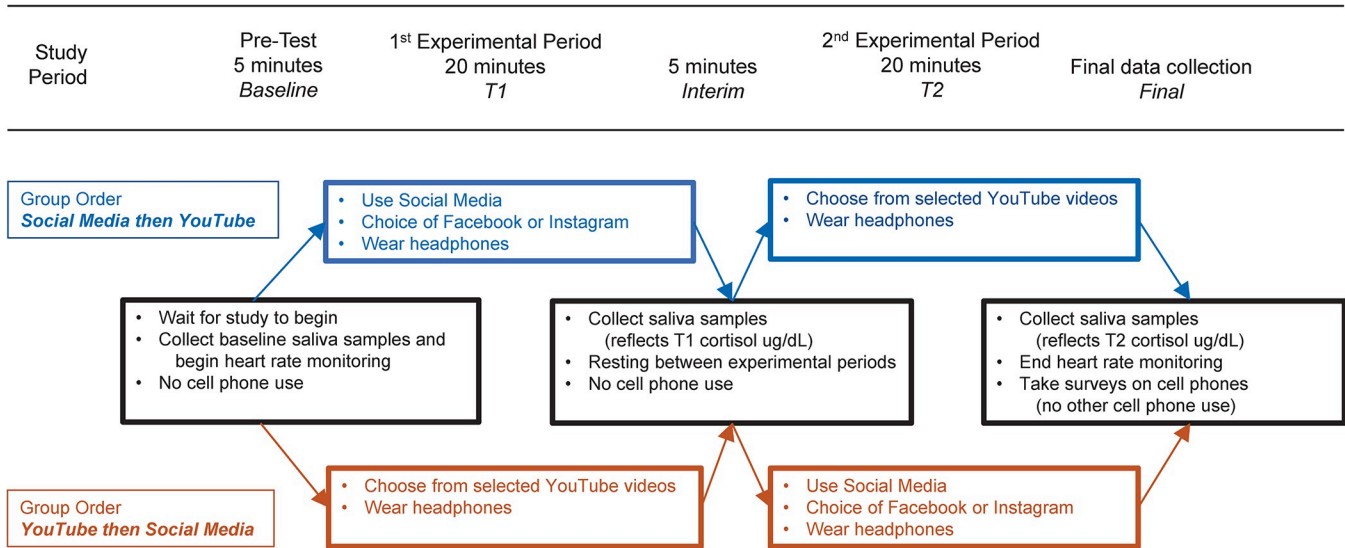

**Fig 1. Schematic of study design.** Study periods are Baseline (the five minutes prior to the first cell phone activity), T1 and T2 identify the two 20-minute periods during which cell phones were used for Social Media or YouTube, Interim identifies the five-minute period between the two 20-minute periods, and Final reflects the period of the final saliva collection and when participants filled out the survey. The saliva samples collected in the Interim and Final periods reflect the response from the first and second 20-minute periods, respectively.

Participants were asked to attend a single experimental session on one of three testing days at 12:30PM and to refrain from eating at least one hour prior to testing. The standardization of time across sessions was important to avoid the influence of the cortisol circadian rhythm [42] and participants were asked to refrain from eating and drinking because some foods and alcohol can compromise performance of the assay used [41]. Participants were assigned a random number, and all collected data were linked to the participants' numbers and not their names or other identifiers. Subjects signed informed consent forms and were randomized into one of two groups.

We employed a two-period crossover design [43,44] to evaluate the physiological responses to social media (experimental treatment) and a non-evocative YouTube playlist (control treatment). With this approach, all participants experienced both treatments but the order in which they experienced them was randomized (Social Media first or YouTube playlist first). The two orders allow clarification of a **treatment** effect (Social Media vs. YouTube), in addition to an **order** effect (Social Media first vs YouTube playlist first) and a **period** effect (first 20-minute period vs. the second 20-minute period, regardless of treatment). Crossover designs are commonly used in medicine [45,46], but have also been used in wildlife [47] and metagenomics [48] studies. They are effective for comparing treatments within the same subject because they differentiate the treatment effect from the period, order, and possibly carryover effects (e.g., the possibility that responses in the second 20-minute period are influenced by responses in the first 20-minute period).

On the day of testing, participants entered the testing room and were fitted with arm-band heart rate monitors by our research team. Heart rate monitors continuously collected data until the conclusion of our protocol. During the pre-test 'Baseline' Period, participants sat quietly (without interacting with their cell phones or each other) for 5–15 minutes (depending on order of entry) and waited for instructions (Fig 1). Subjects were briefed on the study protocol and baseline saliva samples were collected. On each of the three testing days, participants were sampled simultaneously; heart rate recordings and cortisol samples were collected at the same

time, and cell phone treatments (while on individuals' cell phones) were administered simultaneously to the entire group. Among the three testing days, sampling times varied by up to eight minutes.

During the first 20-minute period, "T1," participants were asked to use their personal cell phones to view either a non-evocative YouTube playlist (*YouTube* treatment) or to browse their Facebook or Instagram accounts (*Social Media* treatment). Participants wore headphones during cell phone activities to avoid distracting one another. Because we were interested in the effect of participants' typical social media experiences, rather than the effect of a particular platform, we asked participants to select their customary social media platforms from among the two most popular: Instagram and Facebook [49]. The non-evocative YouTube playlist was used as a negative control and consisted of a selection of videos (e.g., *Neutral Emotion Pictures Video* and *Things You Did Not Know the Use Of*) from which participants could select content to view (a link to this playlist and a list of and links to videos found within this playlist can be found in S1 Appendix); we chose to use this as a negative control because when interacting with this playlist, participants engaged in a similar scrolling and viewing behavior as when selecting content on their social media accounts. This cell phone activity lasted 20 minutes and was followed by a 5-minute interim period. This five-minute interim period encompassed the time that was required to collect the saliva samples; the cortisol measurement taken during the interim period reflects the cortisol response during the first 20-minute period. Participants did not interact with their cell phones or each other during this interim period.

During the second 20-minute period, "T2," subjects were again asked to use their personal cell phones for 20 minutes to view either *YouTube* or *Social Media*; this selection alternated from the subject's first treatment, so that each subject completed both treatments but in random order. Following this 20-minute cell phone activity, a third saliva sample ("Final") was collected, after which participants completed an online demographic and psychological survey via google forms (see Psychological measures). Surveys were administered at the end of our data collection to avoid the possibility that the surveys would themselves cause stress and influence our results. We chose 20-minute experimental periods because several studies have shown that salivary cortisol peaks 20 minutes after a psychological stressor [50–52]. Heart rate monitors were collected from participants after they completed the questionnaires. Fig 1 provides a schematic of the overall study design, indicating the study periods and the identifying terms used throughout this article.

## Physiological measures

To evaluate subjects' physiological responses, both heart rate and salivary cortisol levels were measured. Subjects wore Polar OH1 (Polar Electro Oy, Finland) optical heart rate sensors around their upper arm to continuously record heart rate for the duration of the experiment. These sensors utilize photoplethysmography technology to determine heart rate and have a sampling rate of 50Hz.

Using time records, heart rate data were encoded to match the study periods described in Fig 1. We calculated the average heart rate from the final five minutes prior to T1 and T2 and identified them as the Baseline and Interim periods, respectively. For T1 and T2, we used the average heart rate from the full 20 minutes of each period. Unreported comparisons based on the last five minutes of all periods yielded no change in the results.

Subjects provided saliva samples before the first cell phone treatment and immediately after each of the two treatments. To collect saliva, subjects placed absorbent swabs (SalivaBio Oral Swabs, Salimetrics; State College, PA) in their mouths for 2 minutes and then removed and sealed the swabs in purposely designed tubes. Swabs were centrifuged and the resulting saliva

samples (approximately 1ml) were frozen at -20˚C for up to one week before analysis. Salivary free cortisol was measured using a commercially available enzyme immunoassay kit (Salivary Cortisol ELISA Kit, Salimetrics) [41].

All samples, standards, and controls were run in duplicate, and the manufacturer's protocol was followed without modification. A standard curve was run with each assay, as were high and low standards (provided with the kit). Briefly, cortisol in the provided standards and in our unknown samples competed with cortisol conjugated to horseradish peroxidase for binding sites on capture antibodies (bound to the microtiter plate). After an incubation and wash step, tetramethylbenzidine was added and changed color when it reacted with the horseradish peroxidase (conjugated to cortisol). The amount of color change, quantified by a plate reader (Bio-Rad, iMark Microplate Absorbance Reader; Hercules, CA), was inversely proportional to the amount of cortisol present in the unknown samples. Data reduction software provided with the plate reader was used to interpolate our data using a 4-parameter non-linear regression curve fit. The lower limit of sensitivity for this assay was 0.007μg/dl, and the inter-assay variation was 5.6% and 5.4% for high and low controls, respectively. Intra-assay variation ranged from 2.3–5.4%. All indicate acceptable limits of variation. The cortisol antiserum cross-reacts with dexamethasone (19.2%), but all other cross-reactivity values were less than 0.6%. To stabilize the variance, the base 10 logarithm of cortisol was used in data analysis [53].

## Psychological measures

Because psychological factors can influence one's physiological response [54], we surveyed participants about their social media use, self-esteem, tendency to make social comparisons, and their ongoing stress, using validated and widely used assessments (Table 2). These factors were examined because they could alter the relationship between the physiological stress response and cell phone activities in this study. Assessments were self-reported with personal cell phones via Google Forms, following saliva sampling in the "Final" period (see Fig 1), and scale outcomes were calculated as defined by their developers. To avoid the possibility that thoughts of stress prompted by the Perceived Stress Scale could influence responses on other scales, this was the last scale to be administered. Scales are briefly described in Table 2, and additional details can be found in the S2 Appendix.

## Statistical analysis

Using 2-group t-tests, we confirmed equality of the two groups (*Social Media then YouTube* and *YouTube then Social Media*) for age, psychological measures, and baseline measures of heart rate and cortisol. We confirmed our groups were similar in terms of sex and social media

**Table 2. Validated psychological scales used in our study to quantify psychosocial factors.**

| Psychological Scale | Measures: | Reference |
|---|---|---|
| Intensity of Social Media Use (Facebook Intensity Scale; Instagram Intensity Scale) | Participants' engagement with the platforms with respect to their number of "friends," daily time engagement, emotional investment, and extent to which one's daily activities are integrated with the platform | [55,56] |
| Self-Esteem Scale | Feelings of self-worth and self-acceptance | [57] |
| Social Comparison Scale | Self-perception of social rank, relative attractiveness, and acceptance within one's social group, as pertains to social media | [56,58] |
| Perceived Stress Scale | Self-assessed feelings of stress | [59] |

Additional details in the S2 Appendix.

preference using chi-square tests. Because we randomly assigned individuals to group order, this assessment was necessary to ensure group equality on these key elements.

We confirmed a lack of carryover from the first 20-minute period to the second following standard protocol [43,44]. We fit a repeated measures model in a mixed model framework, incorporating fixed factors of treatment group (order), period, their interaction, and random effect of participant. We then added each potential moderator (sex, age, and the four psychological measures) to the model to examine whether there was a change in the observed relationship between the treatments and physiological stress response; moderators were added independently and were not combined into a single model. We used self-reported sex and discretized age as under 25 years ("digital natives" who have grown up with social media) and 25 years or older. In the event of statistical significance in the 3-way interaction, we used model-based estimates of the outcomes at selected values of the moderator to better understand these interactions. Based on the model results (indicating a significant effect of period), we explicitly tested for a treatment effect on heart rate and cortisol using only the first 20-minute period in a two-group t-test. We adjusted for multiplicity using the false discovery rate (FDR) adjustment [60].

All statistical models used were evaluated for appropriateness with residual plots, including whether there were differences in mean response or variance among the study days. Statistical significance was defined as p (or adjusted p) $\leq 0.05$. Linear modeling was completed with SAS 9.4 and all other descriptive statistics and plotting were completed in R version 4.0.2 [61] using the *tidyverse* library [62] in the RStudio IDE [63].

## Results

We first verified that our two groups (Social Media then YouTube and YouTube then Social Media) were equivalent with respect to survey measures and demographics (Table 3). We noted that Facebook Intensity was slightly higher in the Social Media than YouTube group. Men and women were equally likely to use Facebook or Instagram ($\chi^2_1 = 0.240$, p = 0.624), but on average those using Facebook were older than those using Instagram regardless of group ($T_{36.2} = 4.36$, p<0.001; Facebook mean age 33.6 (2.5), Instagram mean age 21.5 (1.2)).

Carryover effects did not significantly differ for the two group orders for either heart rate ($F_{1,50} = 0.02$, p = 0.878) or cortisol ($F_{1,51} = 1.21$, p = 0.277). Heart rate and cortisol measures by

**Table 3. Summary statistics of key demographics and test of equivalence.**

| Variable | Social Media then YouTube N = 29 | YouTube then Social Media N = 30 | Test of group equivalence |
|---|---|---|---|
| Age | 29.4 (2.3) | 24.4 (1.9) | $T_{54.6} = 1.7$, p = 0.092 |
| Proportion Female | 0.7 (0.1) | 0.7 (0.1) | $X^2_1 = 0.0$, p = 0.928 |
| Proportion Facebook Users | 0.4 (0.1) | 0.5 (0.1) | $X^2_1 = 0.0$, p = 0.883 |
| Proportion Instagram Users | 0.6 (0.1) | 0.5 (0.1) | |
| Social Comparison | 57.7 (2.1) | 61.0 (2.4) | $T_{56.2} = -1.0$, p = 0.310 |
| Perceived Stress | 39.9 (1.2) | 39.2 (1.3) | $T_{56.8} = 0.4$, p = 0.698 |
| Self Esteem | 21.1 (0.8) | 20.3 (0.8) | $T_{57.0} = 0.7$, p = 0.464 |
| Social Media Intensity | 24.7 (0.9) | 23.1 (1.1) | $T_{55.9} = 1.1$, p = 0.260 |
| FB Intensity | 26.5 (1.0) (n = 12) | 22.7 (1.5) (n = 14) | $T_{21.6} = 2.1$, p = 0.052 |
| IG Intensity | 23.4 (1.3) (n = 17) | 23.4 (1.5) (n = 16) | $T_{30.1} = 0.0$, p = 0.991 |

Value shown for each group is mean (standard error).

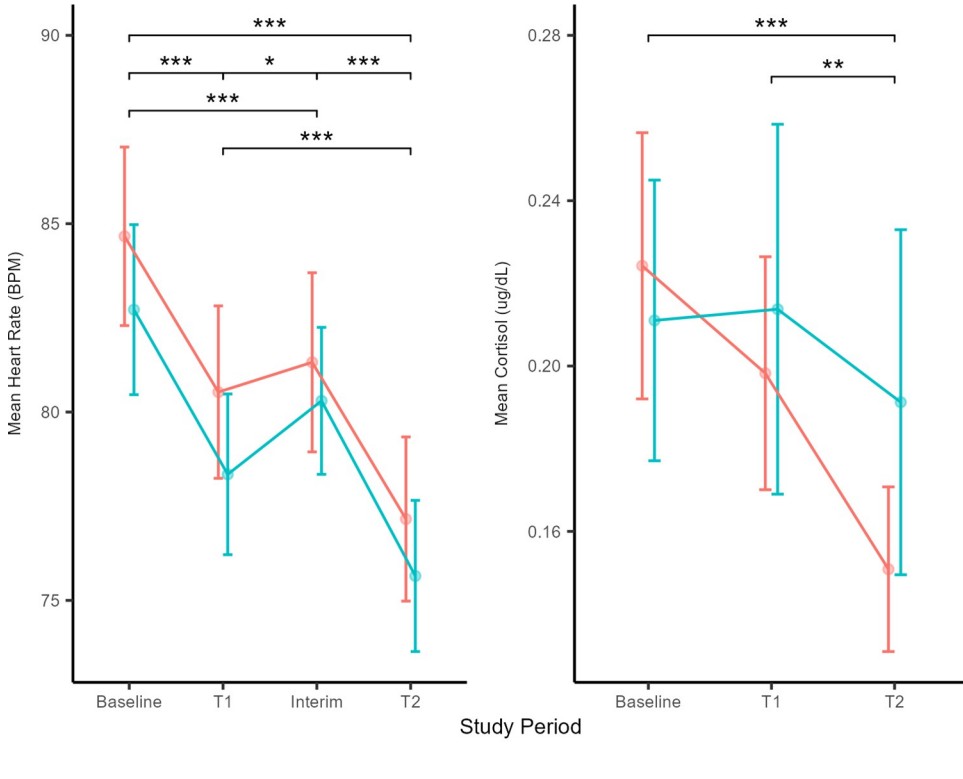

**Fig 2. Summary of physiological measures by study period.** Left panel shows mean heart rate (BPM) ± standard error of the mean; right panel shows mean cortisol (μg/dL) ± standard error of the mean. Raw cortisol values are plotted and reflect titers at baseline and during the two 20-minute treatment periods (T1 and T2); the inference is made from the mixed model using log10(Cortisol) values. Because there was no significant effect of Treatment Order or Treatment Order X time, comparisons of period are based on the main effect of period in the mixed model. Bars identify significantly different periods, *** indicates $p_{adj} < 0.001$, ** indicates $p_{adj} < 0.01$, and * indicates $p_{adj} < 0.05$. Adjustment using FDR [60].

study period are displayed in Fig 2. At baseline, heart rate was equivalent between groups ($T_{51}$ = 0.59, p = 0.555) as was cortisol ($T_{54}$ = 0.63, p = 0.53).

The period effect was statistically significant for both heart rate and cortisol in the linear models (Table 4), and the decrease in physiological response over the course of the study was not different between the two group orders (non-significant group order by period interaction, Table 4). Heart rate differed in all four periods (Fig 2), declining from Baseline to T1 and from Interim to T2, with a slight increase from T1 to Interim. This occurred regardless of group order. Heart rate during the Interim period was lower than during Baseline (Fig 2). From Baseline to T2, heart rate dropped an average of 7.3 (SE = 0.65) beats per minute (BPM). Cortisol concentrations were significantly lower following the second cell phone period (T2) than in the first two study periods (Fig 2). Participants' heart rates and cortisol levels ultimately decreased across the duration of the study, regardless of the cell phone treatments.

Considering the statistically significant period effect and non-significant interaction between treatment order and period, we directly compared heart rate and cortisol responses between cell phone treatments during T1 only. There were no significant differences in mean heart rate ($T_{153}$ = -0.22, p = 0.827) or cortisol ($T_{106}$ = 0.40, p = 0.687) between those viewing Social Media and YouTube during the T1 period.

**Table 4. Analysis of variance results for study outcomes.**

| | Mean Heart Rate (BPM) | | | Cortisol (ug/dL) | | |
|---|---|---|---|---|---|---|
| **Base Model** | | | | | | |
| Fixed Effects | DF (num, den) | F-Value | p-Value | DF (num, den) | F-Value | p-Value |
| Treatment order | 1, 51 | 0.3 | 0.587 | 1, 53.7 | 0.00 | 0.972 |
| 20-minute period | 3, 153 | 62.54 | < .0001 | 2, 105.9 | 9.56 | < .001 |
| Order*Period | 3, 153 | 0.45 | 0.720 | 2, 105.9 | 2.17 | 0.119 |
| Random Effects | | | Estimate | | | Estimate |
| Within participant | | | 121.06 | | | 0.06 |
| Residual | | | 7.76 | | | 0.02 |

We separately added sex and age to our models examining the effects of order and period (as indicated by a 3-way interaction) and found that neither heart rate nor cortisol responses were moderated by participants' sex (heart rate $p = 0.361$, cortisol $p = 0.442$) or age (heart rate = 0.545, cortisol $p = 0.837$). Next, we considered the moderating effects of social media platform (Facebook or Instagram) and psychological measures (as described in Table 2) on heart rate and cortisol measurements. We found only a single significant 3-way interaction when examining the moderating effect of Intensity of Social Media Use on cortisol ($F_{2,100} = 3.15$, $p = 0.047$). High Intensity of Social Media use was associated with elevated cortisol at the baseline experimental period among those in the *Social Media then YouTube* group. As this only affects measurements before the experiment occurs, and doesn't reflect our experimental conditions, we do not believe it is relevant to our study conclusions and do not discuss it further.

## Data availability statement

Accession numbers and/or DOIs will be made available after acceptance and title and abstract have been accepted by reviewers.

## Discussion

Because of concerns surrounding the pervasive use of social media, we set out to examine whether social media provokes a physiological stress response. The crossover design of our experiment allowed us to differentiate the effect of our social media treatment from the effect of treatment period, and we found a period effect but no treatment effect. Among those participating in our study, 20 minutes of social media use did not elicit a physiological stress response and instead, the use of smartphones seemed to reduce activity of the sympathetic nervous system and HPA axis. Further, despite evidence of sex and age-specific differences in digital media use and psychological well-being [64], we found no evidence of age or sex modifying the physiological responses to our smartphone treatments. Additionally, neither psychological traits nor habits of social media use modified the physiological responses in our experiment. We were surprised by these findings because a growing body of literature suggests that excessive smartphone use, and social media use in particular, is detrimental to mental and physical health [10,16,65]. We expected to find that social media use would trigger a measurable physiological stress response and that this might help explain its contribution to psychiatric and physical morbidities. Instead, our findings seem to lend credence to the suggestion [36,37] that accessing social media (and in the case of our study, YouTube) acts to assuage stress.

Nevertheless, despite the absence of a physiological stress response to our social media treatment, our findings do not preclude the possibility that social media use induces psychological stress as has been frequently posited [66–68]. Among studies examining the physiological response to psychological stressors, some have elicited robust physiological responses [69–71], while other studies fail to find an association [26,72]. These contradictory findings may be explained by variations in the cognitive appraisal of a stressor; some perceptions of stress are more likely to elicit a physiological response than others [70,73,74]. Specifically, stressors that are appraised as *uncontrollable*, *novel*, *challenging*, and/or *threatening* are most likely to activate the HPA axis and sympathetic division of the autonomic nervous system [73]. When stressors are appraised in this way, the situation may be perceived as requiring extra resources and thus physiological and behavioral modification are initiated to help provide these resources. In reference to the findings in our study, content viewed for 20 minutes on smartphones may not have been appraised as uncontrollable, novel, challenging, or threatening and thus did not provoke a physiological stress response.

Mobile devices are *controllable* and are not *novel*. While cell phone alerts solicit attention, users can control the amount and type of information disseminated in their feeds. Further, cell phones have lost their novelty and are ubiquitous in our culture. Perhaps when cell phones were novel their use prompted a physiological response, but more recently this response has become habituated, such that a diminished physiological response occurs with repeated exposure to the same stimulus [75,76]. Our participants may have been habituated individuals using their cell phones in the manner to which they have become accustomed. Consequently, a physiological stress response was not provoked by our experimental treatment.

When we gave our study instructions, fitted heart rate monitors, and collected saliva samples, we may have inadvertently introduced a sense of novelty and uncontrollability, which has been shown to elicit a physiological stress response [77–79]. This could explain the relatively higher heart rate during the Baseline and Interim study periods than during the cell phone treatment periods; the sympathetic nervous system seemed to be more active during periods of smartphone abstinence than during periods of smartphone use. Similar findings have been reported by Johnshoy et al. [37] and Rus and Tiemensma [36] who found that social media use on personal devices attenuated the physiological stress response to an acute stressor (Trier Social Stress Test). It may be that contact with smartphones offers a form of social support, even if an individual does not interact with members of one's social group through social media. Access to the phone itself may be a reassuring comfort in times of uncertainty.

Social media and YouTube may not have been perceived as *challenging* or *threatening*. While we did not directly survey feelings of challenge or threat, we surmise our social comparison and self-esteem scales offer insight into such emotions. Those who consistently rank themselves as inferior on the social comparison scale may have felt threatened by online interactions with their "friends," and those who ranked themselves as superior might have felt challenged. Similarly, it can be assumed that those with low self-esteem may have felt more challenged or threatened by social media than those with higher self-esteem. Studies have suggested that threats to the social self (i.e. situations that could lead to rejection of an individual's self-worth) often result in psychobiological responses, including HPA axis and cardiovascular activation [70,71]. As such, we expected to find a heightened physiological stress response when individuals in our study viewed social media. However, we found no relationships between physiological measures of stress and the responses to self-esteem or social comparison scales.

Psychological stressors are first processed by higher brain centers before activating physiological responses [80]. Further studies may attempt to elucidate if higher-order processing results in the social media and YouTube content being appraised as benign; posts that could be

construed as either threatening or challenging may undergo refinement by cognitive processes so that they are appraised as harmless. It may not be social media, per se, that evokes a response but rather the perception of social media content that determines the physiological response. Further, social media users may "prune their feed" such that threatening or challenging posts are entirely eliminated from view, but our research did not examine this possibility.

Because cell phones and social media have only recently come into widespread use, much is left to learn about the social, psychological, and physiological implications of this novel technology. Previous studies have examined social media use with various methodologies [e.g., 13,20,38], and our study is unique in that it evaluated the physiological consequence of social media use using a robust cross-over design. Our findings add to a growing body of knowledge and imply that short-term bouts of cell phone use may not stimulate an acute physiological stress response.

## Limitations and future directions

As our study population reflected a subpopulation of individuals who consented to engage with their social media accounts and smartphones for an hour, this group may not be representative of the general population. Also, we did not control the social media content viewed by any of the participants. While we maintain that our study adequately assessed responses to participants' *typical use* of social media, we acknowledge that our design, which allowed participants to view their preferred content, may have introduced some uncontrolled variability to our study. In addition, it is possible that smartphone use does induce a physiological response but that it might be delayed or only noticeable with extended bouts of use. Our study, in which we examined only the first 20 minutes of use, did not assess this possibility due to participant burden.

Beyond excluding those who ate within one hour of our study from cortisol analyses, we did not control for food or beverage consumption or emotional state (e.g., caffeine consumption, hard exercise, exogenous stressors, or calming activities), prior to collecting our data. It is possible that such events could have altered participants' responses to cell phone treatments throughout the duration of our measurements, and our analyses could not have controlled for this variation in response. However, even if such uncontrolled variables had an effect, it would not change our conclusion. If social media did provoke a stress response, then we would have seen either no change in or an increase in heart rate and cortisol across the duration of our study period and not the observed decrease in these biomarkers.

Lastly, because we used a convenience sample of participants, we may have had an inadequate number of teenagers and older adults participating, which may skew our results and conclusions. To adequately test the moderating effects of sex and age, a larger sample size across both variables may be required.

Our study does not preclude the possibility that social media use can be harmful. Several studies have suggested that social media use can promote negative emotions and distress [81–83], and digital technologies (including social media) have been shown to displace beneficial activities [84], such as school work [85] and exercise [65]. We did not measure such potential negative outcomes. In light of the current online environment, in which divisive discourse and extremist views are commonplace, it may be important to examine the emotional and behavioral responses to social media activities even if they do not increase heart rate and cortisol levels.

## Conclusions

While it may be prudent to consider the physiological and psychological effects of social media, the dangers of new technologies are often overblown. In the Victorian age, print

publications were thought to distort human learning and communications, and the wireless telegraph was feared to isolate users from human interactions [86]. The results of our study add to a small body of evidence suggesting that social media use may in fact serve to mitigate stress [36,37]. Recognizing the limitations of this study, we found that 20 minutes of social media use among our study participants did not provoke a physiological stress response but instead that sitting still and scrolling through content on smart phones likely depressed activity of both the sympathetic nervous system (as evidenced by a decreased heart rate) and the HPA axis (as evidenced by decreased cortisol). More research is needed, to disentangle the physiological stress response from the psychological stress response, and emotional and behavioral responses in general, to more intensive periods of social media use.

## Supporting information

**S1 Appendix. YouTube videos included in our non-evocative playlist.**
(DOCX)

**S2 Appendix. Psychological scales.**
(DOCX)

## Acknowledgments

This study was conducted as part of a summer research experience for undergraduates who participated in the INBRE Summer Scholars program at College of Western Idaho. We are indebted to the following students for their help with and dedication to this project: Brittany Beers, Zoey Carr, Elizabeth Carter, Cristiana Holmes, Abdi Mohamed, Brandi Sweet, Riley Woodworth.

Special thanks to Michelle Fellows for her support in early concept development and Devaleena Pradhan and Keith Sockman for helpful comments on the manuscript.

## Author Contributions

**Conceptualization:** Suzanne Oppenheimer, Laura Bond, Charity Smith.

**Data curation:** Suzanne Oppenheimer, Laura Bond, Charity Smith.

**Formal analysis:** Suzanne Oppenheimer, Laura Bond, Charity Smith.

**Funding acquisition:** Suzanne Oppenheimer, Laura Bond.

**Investigation:** Suzanne Oppenheimer, Charity Smith.

**Methodology:** Suzanne Oppenheimer, Laura Bond, Charity Smith.

**Project administration:** Suzanne Oppenheimer.

**Resources:** Suzanne Oppenheimer.

**Software:** Suzanne Oppenheimer, Laura Bond.

**Supervision:** Suzanne Oppenheimer.

**Validation:** Suzanne Oppenheimer, Laura Bond.

**Visualization:** Suzanne Oppenheimer, Laura Bond, Charity Smith.

**Writing – original draft:** Suzanne Oppenheimer, Laura Bond, Charity Smith.

**Writing – review & editing:** Suzanne Oppenheimer, Laura Bond, Charity Smith.

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
