## [Decision Letter · Decision Letter 0]

17 Jul 2023

PONE-D-23-15507Routine Use of Social Media Does Not Provoke a Physiological Stress ResponsePLOS ONE

Dear Dr. Oppenheimer,

Thank you for submitting your manuscript to PLOS ONE. After careful consideration, we feel that it has merit but does not fully meet PLOS ONE’s publication criteria as it currently stands. Therefore, we invite you to submit a revised version of the manuscript that addresses the points raised during the review process.

ACADEMIC EDITOR: It is undoubtedly an original and therefore promising work. However, it is necessary to make substantial changes, which have been indicated by our reviewers.

We look forward to receiving your revised manuscript.

Kind regards,

Juan-Luis Castillo-Navarrete, Ph.D.

Academic Editor

PLOS ONE

Journal Requirements:

Reviewers' comments:

Reviewer's Responses to Questions

**Comments to the Author**

1. Is the manuscript technically sound, and do the data support the conclusions?

Reviewer #1: Yes

Reviewer #2: Partly

2. Has the statistical analysis been performed appropriately and rigorously? 

Reviewer #1: Yes

Reviewer #2: Yes

3. Have the authors made all data underlying the findings in their manuscript fully available?

Reviewer #1: Yes

Reviewer #2: Yes

4. Is the manuscript presented in an intelligible fashion and written in standard English?

Reviewer #1: No

Reviewer #2: Yes

5. Review Comments to the Author

Reviewer #1: This article addresses the critical and recent topic of the health impact of social media usage. The research aim was to test the increase in participants' stress levels due to the prolonged use of Facebook and Instagram online platforms. The literature has reported controversial results, well described in the paper. Surprisingly, the analysis yielded unexpected results: the stress levels significantly decreased throughout the experiment. Overall, the study is an addition to current knowledge. Ethics, data availability and experimental design planning and implementation are the strengths of this paper. Nonetheless, some crucial adjustments must be considered before it can be accepted and undergo further stages.

1. Terminology and writing: While the study appears sound, the language is unclear, potentially failing to communicate the message. Specifically:

- The authors use the verb "provoke" in the title, abstract, introduction, discussion and conclusions. It appears that the use of the verb "provoke" is inconsistent with the literature that has been introduced (page 1, the articles cited refer to "rising, augmenting and delaying" stress levels, not "causing" as the verb "provoke" suggests) and with the aim of the study (page 2, "Our study directly evaluates the relationship between…").

- In the introduction (page 1), the term "curated" does not align with the meaning intended by the authors (WordReference: From the verb curate. 1. a member of the clergy employed to assist a rector or vicar.)

- The term "routine use" (title and introduction, page 2) may be misused. Although the word "routine" may suggest that the intervention is performed regularly, it is important to note that it is only assessed once in this study. The authors asked participants to use social media as "typical", but the study design was not longitudinal.

- The experiment implementation is described clearly (1 group per day: 15 people on the 1st, 21 on the 2nd, 23 on the 3rd). However, it also says, "On a given day, all participated simultaneously".

-The term "period effect" used by the authors is unclear and requires further explanations. Both experimental conditions had the same duration (page 4). On page 7, the authors write "the course of the study". The results (page 8) report the "period effect", but the sentence is complex and lengthy. It would be beneficial to include a clear explanation earlier in the manuscript rather than at the end of the Results section (page 8, "Participants' heart rates and cortisol levels ultimately decreased across the duration of the study, regardless of the cell phone treatments.").

- Page 9 reports "no significant differences in mean heart rate (T153 = -0.22, p = 0.827) or cortisol (T106 = 0.40, p = 0.687) between those viewing Social Media and YouTube during the T1 period.". The sentence contradicts the previously mentioned results.

2. Organisation of the report content: the authors should reorganise the following section to avoid confusion while reading and understanding how the study was implemented.

- In the abstract, essential methodological choices are left out.

- The study rationale is well-explained by the literature background. However, the study hypothesis and expectations are unclear (page 2). Despite it, the aim of the experiment is clearly stated.

- The ethics are in line with the expected standards. However, the participants' paragraph also describes info that should be part of the experiment procedure (page 3).

-The sentence on page 3, "We employed a two-period crossover design (Jones & Kenward 2015, Millikin & Johnson 2009) to evaluate the physiological response to social media relative to non-evocative content on YouTube." It should be better explained.

- The authors do not explain why they chose a you-tube playlist as a control (page 4).

- The methods section does not explain why the authors used salivary cortisol and heart rate as a sign of activation of the sympathetic system and HPA axis and as a physiological indicator of stress (briefly provided in Discussion, page 10).

- The use of objective and self-report measures is a strength of this experiment. However, the manuscript does not mention which arm the heart rate monitor was placed on (and it could impact the accuracy of data collection) (page 3).

- The study included a survey at the end of the experiment (page 4). Since there are divergent approaches and opinions on this, it could be helpful to justify this methodological choice (the only explanation is given at the end of the paper, in Appendix A, page 19).

- On page 6, The authors should clarify their choices regarding the physiological measures. They are essential info to support the interpretation of the data.

- As far as my knowledge goes, the statistical analysis appeared appropriate to the study objective. However, on page 7, the authors write, "Because of the period effect", as if the readers already know about this effect. It was only mentioned in the abstract.

3. Limitations and future perspectives: Throughout the manuscript, the authors have listed a series of limitations (e.g., page 3 Participants, page 11 Discussion, page 19 Appendix A) that should be reported in this paragraph. Also, considering the fascinating findings and topic of the study, the list of "future research perspectives" (page 12) could be more detailed (e.g., page 9 Results; page 11 Discussion).

4. Citations and References:

- As the PLOS ONE webpage "Submission Guidelines" states, this should be titled "Reference list" (instead of "Literature cited") and citations should be listed in Vancouver style.

- The authors report several inferences on psychological outcomes (Page 11, self-esteem and social comparison related to feelings of challenge and threat; Facebook and Instagram perceived as benign; threatening posts undergoing cognitive process so they would feel harmless; page 12, social media use as coping mechanisms; activation of the parasympathetic system caused by "sitting still and scrolling through content"). However, the study did not include or measure these parameters. Because unsupported by the measurements or data collected, it would be safe to accompany those assumptions with adequate literature appropriately cited.

5. Tables & Figures: Overall, figures and tables are easy to interpret and well structured. However, some details may be improved:

- Table 1 (page 2), some words are underlined. The authors should clarify why in the manuscript or the table description.

- The manuscript states that Figure 1 (page 4) reports the "abbreviations used throughout this article". The study design is extensively drawn, but the figure does not include abbreviations.

- Page 5 is blank, and a reason should be provided for this choice.

- The authors describe Figure 2 in the manuscript as reporting "all four periods" (page 8), baseline-T1-interim-T2. However, Figure 2 (page 9) shows the four points in time only for the heart rate data, not the cortisol data. This difference should be clarified.

Reviewer #2: First, it is important to congratulate the authors for the original work submitted. The manuscript addresses an important field of research and a trending topic in the area. The research question and hypothesis made by the authors will fill a gap in the current literature. Authors took a multidisciplinary approach, considering not only physiological responses but also related psychosocial parameters using validated and well-established assessments. Having a more detailed picture of the phenomenon investigated. Statistical approach and analyses performed seemed appropriate and well executed.

However, there are several major and minor aspects that the authors should resolve. The title proposed for the current work is a generalization, it is strongly recommended to change it, specifying the length of the social media use and the physiological stress parameters investigated. It is a bit confusing how the experimental and control condition/treatments are well defined in the research question, but later described both experimental conditions. More clarity and consistency are needed in relation to the use of the following terms: experimental treatment/condition, experimental/treatment/condition order, treatment/condition effect, and period effect. It is not very clear the rationale for using 20-minute treatment periods interspersed by a 5-min interim period. The length of the treatment has not been considered as a limiting factor of the current study. The appropriateness of the chosen control treatment could be debated. An alternative study design, adding a control group (no cell phone) could have been proposed. When assessing the results and future implications of the current study, the length of the treatment used should be always specified in the text. Otherwise, the term "social media use" is quite general, and we don't know if your results can be extrapolated to other treatment durations.

Further full details are provided in the attached review file with comments.

6. PLOS authors have the option to publish the peer review history of their article (what does this mean?). If published, this will include your full peer review and any attached files.

Reviewer #1: **Yes: **Luana Scrivano

Reviewer #2: No

---

## [Author Response · Author response to Decision Letter 0]

5 Sep 2023

We have uploaded files with specific details that address each of the reviewers' comments. Please refer to these documents for our detailed responses.

---

## [Decision Letter · Decision Letter 1]

27 Sep 2023

PONE-D-23-15507R1Typical use of social media does not elicit a short-term physiological stress response as measured by heart rate and salivary cortisolPLOS ONE

Dear Dr. Oppenheimer,

Thank you for submitting your manuscript to PLOS ONE. After careful consideration, we feel that it has merit but does not fully meet PLOS ONE’s publication criteria as it currently stands. Therefore, we invite you to submit a revised version of the manuscript that addresses the points raised during the review process.

The topic is extremely interesting and innovative and there is a noticeable improvement compared to the first submission. However, there are still some important aspects to be corrected, raised not only by the reviewer.

In my opinion, the information obtained from this paper is a significant contribution to the current knowledge, so I strongly urge you to continue with the improvement of the paper. In this regard:

(1) Significantly improve the wording of several paragraphs of the brief.

(2) I suggest incorporating an explanatory figure, such as a flow chart, regarding the inclusion of participants.

(3) It is strongly requested to include information regarding the cortisol assay used, as well as the conditions under which the determinations were made.

(4) In accordance with the above, it is requested to provide more detailed information regarding the interpretation of the cortisol assay results.

(5) For a better understanding, I suggest, as far as possible, to incorporate as supplementary material, a list with links to youtube videos used.

We look forward to receiving your revised manuscript.

Kind regards,

Juan-Luis Castillo-Navarrete, Ph.D.

Academic Editor

PLOS ONE

Reviewers' comments:

Reviewer's Responses to Questions

**Comments to the Author**

1. If the authors have adequately addressed your comments raised in a previous round of review and you feel that this manuscript is now acceptable for publication, you may indicate that here to bypass the “Comments to the Author” section, enter your conflict of interest statement in the “Confidential to Editor” section, and submit your "Accept" recommendation.

Reviewer #1: All comments have been addressed

Reviewer #2: (No Response)

2. Is the manuscript technically sound, and do the data support the conclusions?

Reviewer #1: Yes

Reviewer #2: (No Response)

3. Has the statistical analysis been performed appropriately and rigorously? 

Reviewer #1: Yes

Reviewer #2: (No Response)

4. Have the authors made all data underlying the findings in their manuscript fully available?

Reviewer #1: Yes

Reviewer #2: (No Response)

5. Is the manuscript presented in an intelligible fashion and written in standard English?

Reviewer #1: Yes

Reviewer #2: (No Response)

6. Review Comments to the Author

Reviewer #1: I would like to express my appreciation for the precision with which all the comments and revisions have been addressed. I extend my compliments to the authors for the related adjustments that have been made.

Reviewer #2: Firstly, I would like to congratulate the authors for having answered successfully to most of the questions raised. By doing that, the quality of their work has increased notably, and their effort and professionalism have been shown in the quality of their work. However, there are several concerns about some of the answers (see attached file), particularly in their approach to the concept of statistical significance, study design, and other crucial aspects of a research study, such as addressing limitations.

7. PLOS authors have the option to publish the peer review history of their article (what does this mean?). If published, this will include your full peer review and any attached files.

Reviewer #1: **Yes: **Luana Scrivano

Reviewer #2: No

---

## [Author Response · Author response to Decision Letter 1]

3 Nov 2023

We have addressed specific comments in our document entitled "comments & responses."

---

## [Editor Report · Decision Letter 2]

23 Nov 2023

PONE-D-23-15507R2Social media does not elicit a physiological stress response as measured by heart rate and salivary cortisol over 20-minute sessions of cell phone usePLOS ONE

Dear Dr. Oppenheimer,

Thank you for submitting your manuscript to PLOS ONE. After careful consideration, we feel that it has merit but does not fully meet PLOS ONE’s publication criteria as it currently stands. Therefore, we invite you to submit a revised version of the manuscript that addresses the points raised during the review process.

**ACADEMIC EDITOR: **

Additional Editor Comments:

I would like to highlight the work carried out, which has led to a substantial improvement in the writing. In fucnion of the above, I would like to outline some minor suggestions aimed at enriching the paper.

Abstract

The article abstract provides an adequate overview of the study, which investigates the physiological response to stress, as measured by heart rate and salivary cortisol, in 20-minute sessions of mobile phone use, specifically in the context of social media use and a pre-selected YouTube playlist. While the abstract clearly sets out the scope and key findings of the study, it could benefit from a clearer articulation of its relevance in the broader context of current research on stress and social network use.

State of the Art

The state of the art discussed in the manuscript addresses the existing literature, but could be further elaborated to establish a stronger argument for the need for this research. Given the volume of previous work in the area of social networks and stress, a more detailed comparison with previous studies and a clear justification of how this study contributes to filling a gap in the literature would be beneficial (in my view it would simplify the reading).

Problematisation and Research Question

The problematisation of the study, which investigates whether the use of social networks induces a physiological stress response, is relevant. However, the specific research question could be more clearly formulated to accurately reflect the focus of the study.

Hypotheses and Objectives

The hypothesis that social network use does not elicit a physiological stress response is adequately tested in the study. The aims of the study are clear, although they could benefit from greater specificity in terms of how the results could inform current understanding of the interaction between social network use and stress.

Methodology

The methodology employs a crossover study design with an interesting approach to compare the treatment effect of mobile phone use on the physiological response to stress. However, there are significant limitations:

Sample Selection and Sample Size: The sample of 59 subjects spans a wide range of ages, which is positive, but how differences in age and sex might affect the results could be explored further.

Controlling Variables: The study mentions the lack of control over the content of the social media viewed, which could introduce significant variability in the stress response. In addition, caffeine consumption and other dietary or environmental factors that could influence cortisol and heart rate measures are not adequately controlled for.

Discussion

The discussion provides a detailed analysis of the findings, comparing them with previous studies and related theories. However, the discussion could be strengthened by further addressing how these findings contribute to the existing literature and what implications they have for understanding social media-related stress.

Conclusions

The conclusions are consistent with the results, highlighting the lack of a significant physiological stress response associated with social media use. However, it would be beneficial if the conclusions also addressed the limitations of the study and suggestions for future research (although these have already been discussed).

Statistical Analysis

The statistical analysis, including the use of a crossover design and measures of heart rate and cortisol, is robust. However, interpretation of the statistical results should be cautious, especially given the number of uncontrolled variables and possible variability in participants' individual experience with social networks. I suggest revisiting this point and drafting a paragraph that cautions these concepts.

We look forward to receiving your revised manuscript.

Kind regards,

Juan Luis Castillo-Navarrete, Ph.D.

Academic Editor

PLOS ONE
---

## [Author Response · Author response to Decision Letter 2]

7 Jan 2024

We have included a document titled "Response to Reviewers" that contains altered manuscript text and explanations that address each of the comments in the decision letter.

---

## [Editor Report · Decision Letter 3]

26 Jan 2024

Social media does not elicit a physiological stress response as measured by heart rate and salivary cortisol over 20-minute sessions of cell phone use

PONE-D-23-15507R3

Dear Dr. Suzanne Oppenheimer,

We’re pleased to inform you that your manuscript has been judged scientifically suitable for publication and will be formally accepted for publication once it meets all outstanding technical requirements.

Kind regards,

Juan Luis Castillo-Navarrete, Ph.D.

Academic Editor

PLOS ONE

Additional Editor Comments (optional):

It is undoubtedly pertinent to highlight the work done, which has substantially improved the writing, even more so in a prevalent topic that is undoubtedly only the tip of the iceberg.
---

## [Editor Report · Acceptance letter]

12 Mar 2024

PONE-D-23-15507R3 

PLOS ONE

Dear Dr. Oppenheimer, 

I'm pleased to inform you that your manuscript has been deemed suitable for publication in PLOS ONE. Congratulations! Your manuscript is now being handed over to our production team.

Kind regards, 

on behalf of

Dr. Juan Luis Castillo-Navarrete 

Academic Editor

PLOS ONE